# Atomically precise control of rotational dynamics in charged rare-earth complexes on a metal surface

Tolulope Michael Ajayi [1,2], Vijay Singh[3,4], Kyaw Zin Latt[1], Sanjoy Sarkar [2], Xinyue Cheng [5], Sineth Premarathna [1,2], Naveen K. Dandu [3,4], Shaoze Wang[1,2], Fahimeh Movahedifar[5], Sarah Wieghold [1,6], Nozomi Shirato[1], Volker Rose [6], Larry A. Curtiss[3], Anh T. Ngo[3,4], Eric Masson [5] ✉ & Saw Wai Hla [1,2] ✉

Complexes containing rare-earth ions attract great attention for their technological applications ranging from spintronic devices to quantum information science. While charged rare-earth coordination complexes are ubiquitous in solution, they are challenging to form on materials surfaces that would allow investigations for potential solid-state applications. Here we report formation and atomically precise manipulation of rare-earth complexes on a gold surface. Although they are composed of multiple units held together by electrostatic interactions, the entire complex rotates as a single unit when electrical energy is supplied from a scanning tunneling microscope tip. Despite the hexagonal symmetry of the gold surface, a counterion at the side of the complex guides precise three-fold rotations and 100% control of their rotational directions is achieved using a negative electric field from the scanning probe tip. This work demonstrates that counterions can be used to control dynamics of rare-earth complexes on materials surfaces for quantum and nanomechanical applications.

Rare-earth (RE) elements are tantalizing to explore for advanced technological applications[1-8] because of their distinct magnetic, optical and catalytic properties[9-16]. One of the best options for an efficient usage of RE elements is to place them inside coordination complexes; such systems are uniquely advantageous due to their well-defined and reproducible shapes and sizes. The ligands not only protect the RE ion, but also play vital roles in determining its electronic, magnetic, and optical properties[1,17]. Thus, single molecule studies of charged RE complexes are of great interest to explore the environment of the RE ion that would allow the design of appropriate ligands to tailor their properties. How a charged complex behaves under an external electric field is yet another unexplored question. Here we form rare-earth complexes by coordinating a positively charged Europium base molecule with negatively charged counterions on a Au(111) surface. Electronic and structural properties of these complexes are then investigated on a one complex at-a-time basis using scanning tunneling microscopy (STM) and tunneling spectroscopy methods. We show that an entire RE complex can be rotated like a single molecule[18-27] with precise control of their rotational dynamics at the atomic scale.

[1]Nanoscience & Technology Division, Argonne National laboratory, Lemont, IL 60439, USA. [2]Nanoscale & Quantum Phenomena Institute, and Department of Physics & Astronomy, Ohio University, Athens, OH 45701, USA. [3]Materials Science Division, Argonne National laboratory, Lemont, IL 60439, USA. [4]Chemical Engineering Department, University of Illinois at Chicago, Chicago, IL 60608, USA. [5]Nanoscale & Quantum Phenomena Institute, and Department of Chemistry and Biochemistry, Ohio University, Athens, OH 45701, USA. [6]Advanced Photon Source, Argonne National laboratory, Lemont, IL 60439, USA. ✉e-mail: masson@ohio.edu; hla@ohio.edu

## Results and discussion

### Structures of RE complexes

The base coordination complex described herein is $[Eu(pcam)_3]^{3+}$ (Europium(III) tris(2,6-pyridinedicarboxamide)) unit. This complex was synthesized using a scheme shown in Fig. 1a (see Methods section and Supplementary Method 1 for synthetic details). In solution, $[Eu(pcam)_3]^{3+}$ is composed of three equivalent ligand arms in a planar, distorted $D_{3h}$ geometry with 120° angles between the nearest arms (Fig. 1b). Coordination of a triflate ($CF_3SO_3^-$) counterion on the surface alters the planar geometry of $[Eu(pcam)_3]^{3+}$ to become a nonplanar $[Eu(pcam)_3X]^{2+}$ complex with a distorted trigonal pyramid geometry (Fig. 1c). Density functional theory (DFT) calculations predict that complex $[Eu(pcam)_3X]^{2+}$ binds to the Au(111) surface via the counterion located under the $Eu(pcam)_3$ unit, and the structure (Fig. 1d) remains similar to its gas phase counterpart (Fig. 1c).

We deposited the $Eu(pcam)_3$ triflate salt onto an atomically clean Au(111) substrate using thermal evaporation in an ultrahigh vacuum (UHV) environment. In STM images acquired at ~5 K substrate temperature (Fig. 2a), complex $[Eu(pcam)_3X]^{2+}$ appears as a distorted triangular shape (Fig. 2b) with its three arms positioned on Au(111) along [110] or [211] surface directions. The incorporation of the counterion underneath (Fig. 1b, c, d) induces an upward bending at the centre of $Eu(pcam)_3$, which reduces a side length from 2.1 nm in $[Eu(pcam)_3]^{3+}$ to 1.8 nm in $[Eu(pcam)_3X]^{2+}$ (Supplementary Note 1). In addition, the electronic structure and the shapes of the orbitals are drastically altered in $[Eu(pcam)_3X]^{2+}$ as compared to $[Eu(pcam)_3]^{3+}$ (Supplementary Notes 1, 2, & 3). To make an unambiguous distinction between the complexes with and without counterion underneath, a spectroscopic movie (Supplementary Movie 1) is created from 8000 dI/dV spectroscopic maps acquired over a pair of $[Eu(pcam)_3X]^{2+}$ - $[Eu(pcam)_3]^{3+}$

**Fig. 1 | Structure of rare-earth coordination complexes. a** Synthetic route of pcam (**3**) and $[Eu(pcam)_3](CF_3SO_3)_3$. **b** Top views of $[Eu(pcam)_3]^{3+}$, the triflate anion and complex $[Eu(pcam)_3X]^{2+}$. **c** Side view of complex $[Eu(pcam)_3X]^{2+}$ with the triflate counterion underneath $Eu(pcam)_3$ (circle). **d** A 3-D charge density plot of complex $[Eu(pcam)_3X]^{2+}$ on Au(111) generated by geometrically relaxed DFT calculation.

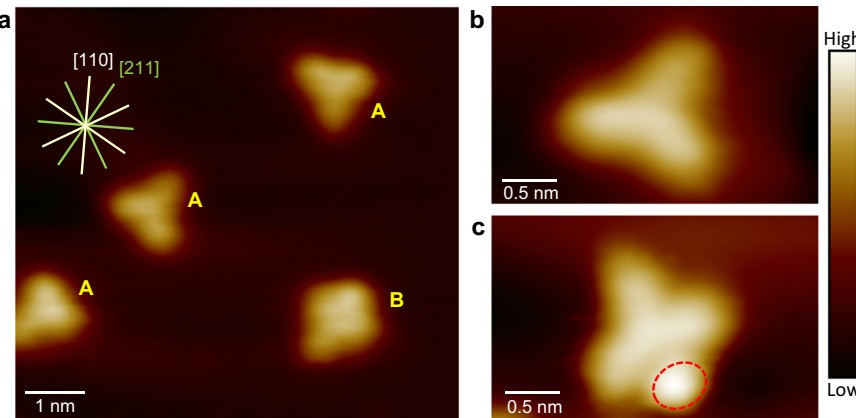

**Fig. 2 | Imaging rare-earth complexes on Au(111). a** An STM image shows individual [Eu(pcam)$_3$X]$^{2+}$ complexes (labeled 'A') and [Eu(pcam)$_3$X$_2$]$^+$ (labeled 'B') on Au(111) [$V_t = -2.0$ V, $I_t = 7.4 \times 10^{-11}$A, 5 K]. **b** A zoom-in STM image of complex [Eu(pcam)$_3$X]$^{2+}$ on Au(111) [$V_t = -2.0$ V, $I_t = 3.5 \times 10^{-11}$A, 5 K]. **c** A zoom-in STM image of complex [Eu(pcam)$_3$X$_2$]$^+$ on Au(111) [$V_t = -1.1$ V, $I_t = 4.3 \times 10^{-11}$A, 5 K]. The dashed circle indicates the additional counterion attached to the side of the complex.

complexes at ±2000 mV range with 1 mV interval between the consecutive frames. This movie clearly reveals different energetic locations and the shapes of unoccupied orbitals of these two complexes between +1200 mV and 2000 mV. We also observe complexes with two counterions [Eu(pcam)$_3$X$_2$]$^+$, where a second, non-coordinating counterion is attached to one side of complex [Eu(pcam)$_3$X]$^{2+}$ (Fig. 2a, c, and Supplementary Note 4). The three arms of this complex are positioned only along [211] surface directions on Au(111). DFT calculations reveal that attaching the counterion to the centre and to the sides of the molecule is favorable in the gas phase as well.

## Electronic and chemical states

The electronic structures of complex [Eu(pcam)$_3$X]$^{2+}$ and the additional triflate counterion on Au(111) are determined by using dI/dV tunneling spectroscopy at 5 K substrate temperature. Tunneling spectroscopy data recorded by positioning the STM tip above the centre of [Eu(pcam)$_3$X]$^{2+}$ (Fig. 3a, b) reveals the energetic positions of the highest occupied and lowest unoccupied molecular orbitals (HOMO and LUMO) as −2 V and +1.2 V, respectively (a HOMO-LUMO energy gap of 3.2 eV). The measured HOMO-LUMO gap agrees with DFT for complex [Eu(pcam)$_3$X$_2$]$^+$ adsorbed on Au(111), where the calculated projected density of states (PDOS) gives the HOMO and LUMO orbitals at −1.8 eV, and +1.1 eV, respectively (Fig. 3c) (Supplementary Note 3). The calculated DOS reveals that the Eu(pcam)$_3$ unit mainly contributes to the HOMO and LUMO (Fig. 3d) while the counterion underneath does not have any electronic states near the Fermi level, i.e., '0 V' (Fig. 3e). Calculations further unveil that only negligible amount of charge transfer is taking place and both complexes remain charged on Au(111) (Supplementary Note 5).

The Eu ion in the [Eu(pcam)$_3$X]$^{2+}$ complex has a +3 oxidation state in the gas phase. If there is no significant charge transfer from the counterions or the substrate to the Eu ion as suggested by the DFT calculations, then it should remain as Eu$^{3+}$ in the complex adsorbed on Au(111). This was further verified by synchrotron X-ray scanning tunneling spectroscopy (SX-STM)[28,29] experiments conducted at the XTIP beamline[30] of Advanced Photon Source and Center for nanoscale Materials at Argonne National Laboratory. During the experiment, a 10 μm × 10 μm monochromatic synchrotron X-ray beam is incident at the tip-sample junction after passing through a high frequency X-ray chopper. A coaxial coated SX-STM tip[31] is used as a detector to collect photoelectrons produced by the X-ray excitations of the sample, and the tip is positioned in the far-field regime[28], i.e., ~5 nm from the sample surface, where no tunneling between the tip and sample is taking place. We then compare a bulk powdered sample placed on a graphite (HOPG) substrate, where its chemical state should closely resemble

that of the gas phase, with a sub-monolayer coverage of complex [Eu(pcam)$_3$X]$^{2+}$ adsorbed on a Au(111) thin-film formed on a mica substrate. The scanning tunneling microscope X-ray absorption spectra (STM-XAS) exhibit cogent $M_5$ and $M_4$ peaks of Eu ion originating from the $3d_{5/2} \rightarrow 4f_{7/2}$ and $3d_{3/2} \rightarrow 4f_{5/2}$ transitions (Fig. 3f) for both samples as expected. The chemical states of the Eu ions in the samples are then determined from the multiplets representing the near-edge X-ray absorption fine structures (STM-NEXAFS)[29]. For the powdered sample, the STM-NEXAFS peaks are observed at 1125.2 eV, and 1133.6 eV, respectively (indicated with arrows in Fig. 3f), and their energetic locations indicate the Eu$^{3+}$ oxidation state[32]. Akin to the powdered sample, similar STM-NEXAFS peaks are observed when a sub-monolayer coverage of the molecular complexes are adsorbed on Au(111) (Fig. 3f) confirming that the complex maintains its Eu$^{3+}$ state as predicted by the theory.

## Rotational dynamics of the complexes

When STM images are acquired at an elevated temperature of ~100 K, complexes [Eu(pcam)$_3$X]$^{2+}$ and [Eu(pcam)$_3$X$_2$]$^+$ appear as a circular shape on Au(111) (Fig. 4a, and Supplementary Fig. S27) caused by random rotations due to thermal excitations. When thermal energy is quenched at a low temperature of 5 K, the complexes become stationary on the Au(111) surface thereby enabling us to selectively rotate them. Stepwise rotations of the complexes are performed at 5 K on a one-complex-at-a-time basis using the electric field emanating from the STM tip[19]. To induce the rotation, the STM tip is positioned at a fixed height above the Eu ion location in the complex (Fig. 4b), and then tunneling bias is ramped from 0 to above ±3 V. When the complex rotates, a sudden change in the tunneling current intensity occurs (Fig. 4c), and the rotation event can be confirmed by acquiring a subsequent STM image[19,21]. The complexes can be rotated either by positive or negative bias (Fig. 4c), and the process is akin to the rotation of rotors on surfaces[20,23] although the complex is composed of multiple components held together just by electrostatic interactions.

Figures 4d–f present a sample sequence of STM images when complex [Eu(pcam)$_3$X]$^{2+}$ is rotated in both clockwise and anticlockwise directions. Here, the rotational axis of the complex is located at its centre. Such a rotation can occur when a part of the complex is attached to the surface, which then can act as a pivot[18] otherwise, the complex will laterally displace by the electric field excitations. The pivot here is the triflate counterion underneath the Eu(pcam)$_3$ unit (Fig. 1c), which acts as a stator while the Eu(pcam)$_3$ unit acts as a rotator. In general, molecular rotors and molecular motors with the sizes of a few nanometers on solid surfaces exhibit quantized rotation behaviors, i.e., they rotate stepwise when energy is supplied[18–22]. From

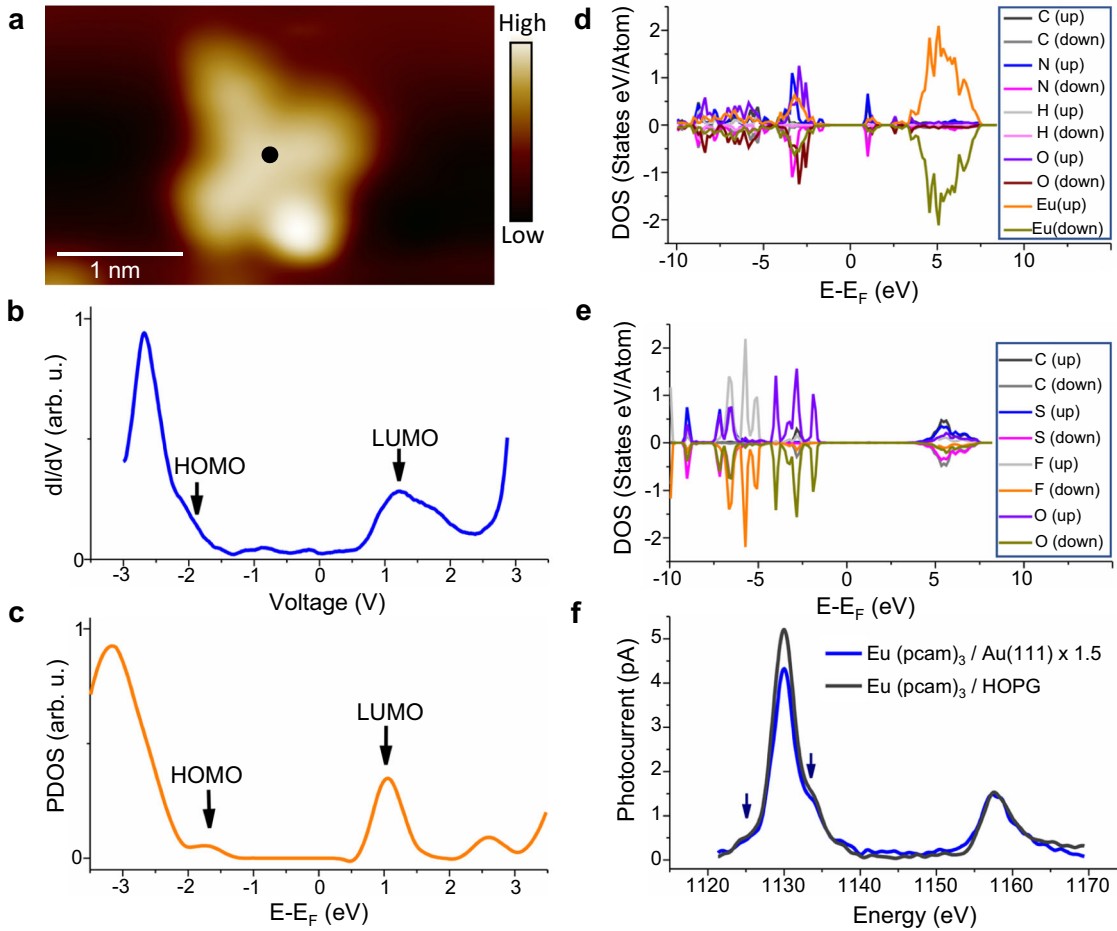

**Fig. 3 | Electronic structure and chemical state. a** STM image of complex [Eu(pcam)$_3$×$_2$]$^+$ on Au(111) [V$_t$ = −1.1 V, I$_t$ = 4.3 × 10$^{-11}$A, 5 K]. **b** A dI/dV spectrum measured at the centre of [Eu(pcam)$_3$X$_2$]$^+$ such as shown with a black dot in (**a**) shows the HOMO, and LUMO. **c** Projected density of states (PDOS) of the complex as a function of energy shows HOMO and LUMO peaks of Eu(pcam)$_3$ unit adsorbed on Au(111). **d** A calculated PDOS of Eu(pcam)$_3$ in the complex adsorbed on Au(111). **e** A calculated PDOS of the triflate counterion underneath Eu(pcam)$_3$ adsorbed on Au(111). **f** STM-XAS spectra of the complex on HOPG and Au(111) showing Eu $M_5$ and $M_4$ peaks. The arrows indicate multiplets showing the Eu$^{3+}$ oxidation state. Source data for (**b**–**f**) are provided as a Source Data File.

the statistical analysis of 102 stepwise rotations, the preferential rotation angle of the complex [Eu(pcam)$_3$X]$^{2+}$ is determined as 60° (Fig. 4g). Similar electric field induced rotations can be performed on complex [Eu(pcam)$_3$X$_2$]$^+$ by positioning the STM tip above the Eu ion location of the complex (Fig. 4h–j). However, surprisingly, the preferential rotation angle collected from 238 stepwise rotations here is found to be 120° (Fig. 4k). Such 120° rotation angle on a surface with a hexagonal atomic lattice is not typical except for small molecules such as O$_2$[33], and C$_2$H$_2$[34] where the absorption sites or surface chirality can dictate their rotations. When the tip is positioned above the Eu ion of the complex during manipulation, both complexes can be rotated in clockwise and anticlockwise directions (Fig. 4d–f, and h–j) with approximately equal probability (Fig. 4l), and thus no directional control of their rotations is achieved so far.

An important parameter to determine is the critical electric field[19] required to rotate the complexes. Rotations of both complexes are performed to that aim by changing the tip height while recording the threshold bias as shown in Fig. 4c. The resulting threshold voltages are plotted as a function of tip height (Fig. 4m). Within the measured range, the threshold voltage linearly increases with the increasing tip height, and the slopes of the plots provide the critical electric fields required for rotation of the complexes. Surprisingly, the measured critical electric field is found to be smaller for complex [Eu(pcam)$_3$X$_2$]$^+$ (1.65 V Å$^{-1}$) than complex [Eu(pcam)$_3$X]$^{2+}$ (2.05 V Å$^{-1}$) (Fig. 4m), which is

not expected because [Eu(pcam)$_3$X$_2$]$^+$ is larger. However, if there is an additional dipole in complex [Eu(pcam)$_3$X$_2$]$^+$, it can change the barrier and consequently alter the critical electric field strength required for rotation.

The triflate counterion has a negative charge (−1) and the DFT calculation shows that it maintains a net charge on Au(111) (Supplementary Note 5). This charged state of the counterion would assist in rotation via Coulomb interactions when an electric field is applied, which reduces the amount of critical electric field required for rotation. To confirm the negative charged state of the side counter ion, we perform rotations by positioning the tip next to the counterion as shown in the STM image sequence of Fig. 5a–c and apply a negative electric field from the STM tip. If the counterion is negatively charged, then the Coulomb repulsion between the tip and the complex would result in pushing it away from the tip. As expected, this procedure always leads to the rotation of the complex in the opposite direction of the tip confirming the negative charge state of the side counterion in the complex. Depending on the tip position left or right side of the side counterion, clockwise or anticlockwise rotation can now be performed at will. To determine a deterministic control over the rotational direction, we have performed 709 controlled stepwise rotations where 100% control is achieved on the rotational direction of complex [Eu(pcam)$_3$X$_2$]$^+$ (Supplementary Note 6, and Supplementary Movies 2, and 3). Such controlled directional rotations can be performed only

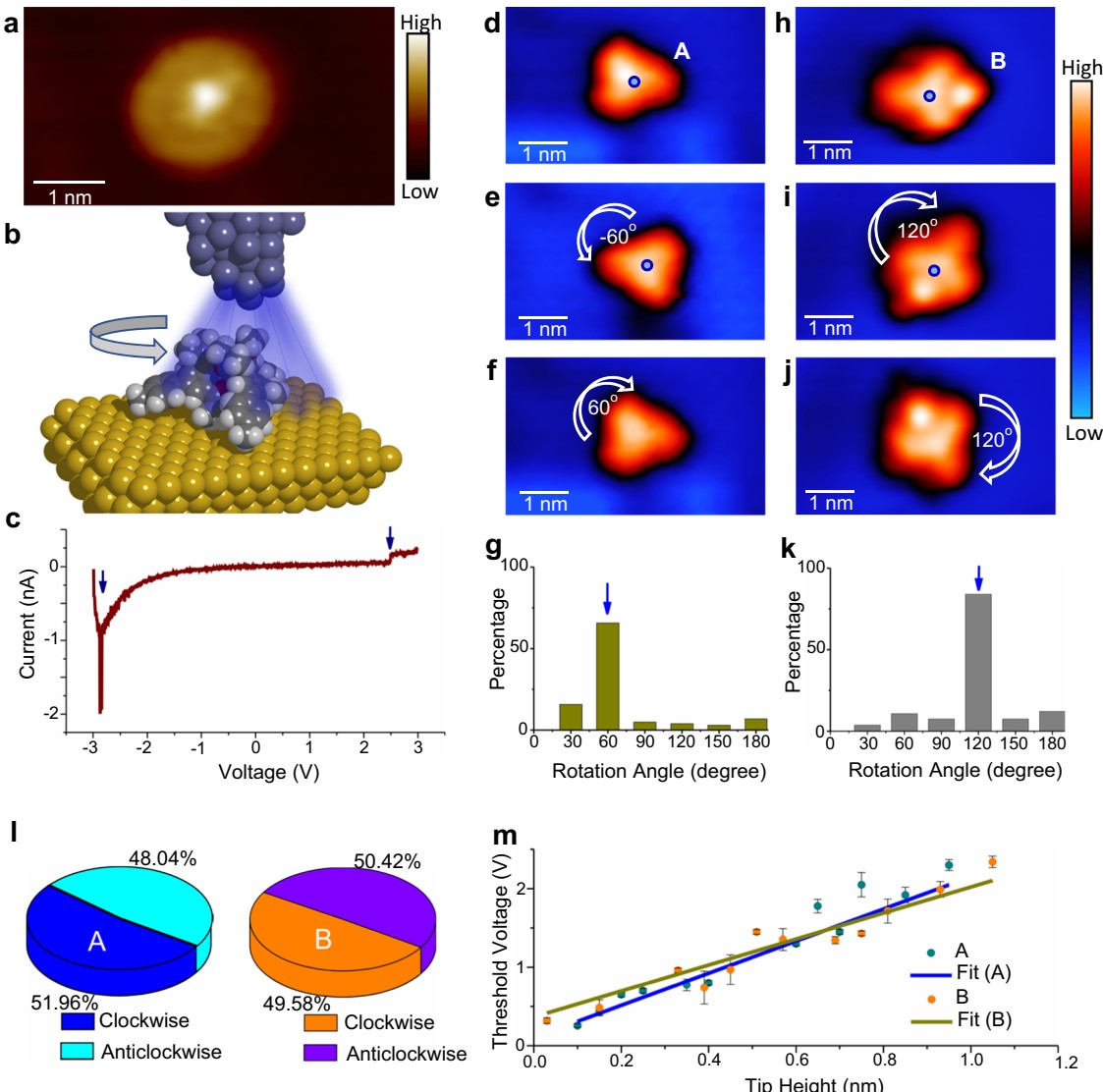

**Fig. 4 | Electric field induced rotations. a** An STM acquired at 100 K shows a rotating complex [$V_t = -1.5$ V, $I_t = 3.0 \times 10^{-11}$A]. **b** Illustration of the STM tip induced electric field rotation process. **c** An I–V spectrum acquired above a complex reveal sudden changes in current (indicated with arrows) due to rotations. Complex [Eu(pcam)$_3$X]$^{2+}$ (labelled "A") in (**d**) is rotated anticlockwise by 60° (**e**) and then clockwise by 60° (**f**) [$V_t = -2.0$ V, $I_t = 3.5 \times 10^{-11}$A, 5 K]. **g** The rotation angle distribution plot shows 60° as the key rotation angle. Complex [Eu(pcam)$_3$X$_2$]$^+$

(labelled "B") in (**h**) is rotated clockwise by 120° (**i**) and then again by 120° (**j**). **k** The rotation angle distribution plot reveals 120° as the key rotation angle for complex [Eu(pcam)$_3$X$_2$]$^+$. **l** The distributions of clockwise and anticlockwise rotations for type A and B complexes. **m** Threshold electric fields for rotations measured as a function of tip height for both complex types. The error bars are from statistical distributions. Source data for (**c**), (**g**), (**k**), (**l**) and (**m**) are provided as a Source Data File.

using the negative electric field from the tip. When the tip is positive, then the complex rotates toward the tip due to Coulomb attraction leading to attachment of the complex to the tip via the counterion.

Next, to understand the 120° rotation angle, we investigate the adsorption geometry of complex [Eu(pcam)$_3$X$_2$]$^+$. As shown in the STM image (Fig. 5d), the pcam arms of [Eu(pcam)$_3$X$_2$]$^+$ always point to [211] directions on the Au(111) surface. The pcam arms direction on Au(111) is indeed dictated by the side counterion (Supplementary Note 7). Figure 5e provides an adsorption site of the complex on Au(111) surface calculated by geometrically relaxed DFT scheme. Here, the side triflate counterion adsorbs on Au(111) surface via its fluorine atoms at the three-fold hollow sites. It is known that fluorine prefers to adsorb at the three-fold hollow sites on Au(111)[35], which is in agreement with our geometrically relaxed calculations for the complex together with the side counterion adsorbed on Au(111) surface as well. The energetic barrier difference between the triflate fluorine atoms adsorbed on the hollow sites and the top sites

is ~0.5 eV. Such preferential adsorption of the added counterion leads to the observed orientation of the complex on Au(111), which imposes only three possible repeating adsorption sites with 120° angles for a 360° rotation (Fig. 5f). Thus, like in the cases of small molecule rotations on surfaces such as O$_2$[33], and C$_2$H$_2$[34], rotation angle of the entire [Eu(pcam)$_3$X$_2$]$^+$complex is now determined by the adsorption site of a small counterion at the side of the complex. By shifting the centre of the complex between fcc and hcp sites on Au(111) the three rotation sites can flip (Fig. 5g). This means that only 120° rotation steps will reproduce the same adsorption condition while 60° rotation would involve lateral translation of the complex by half of the Au atomic distance. Thus, the 120° stepwise rotation angle of the entire complex is now controlled by the side counterion.

Due to the 120° rotation step, each [Eu(pcam)$_3$X$_2$]$^+$ complex does have its mirror image on the surface, and both complexes are non-superposable (a 180° rotation step would allow superposition, not

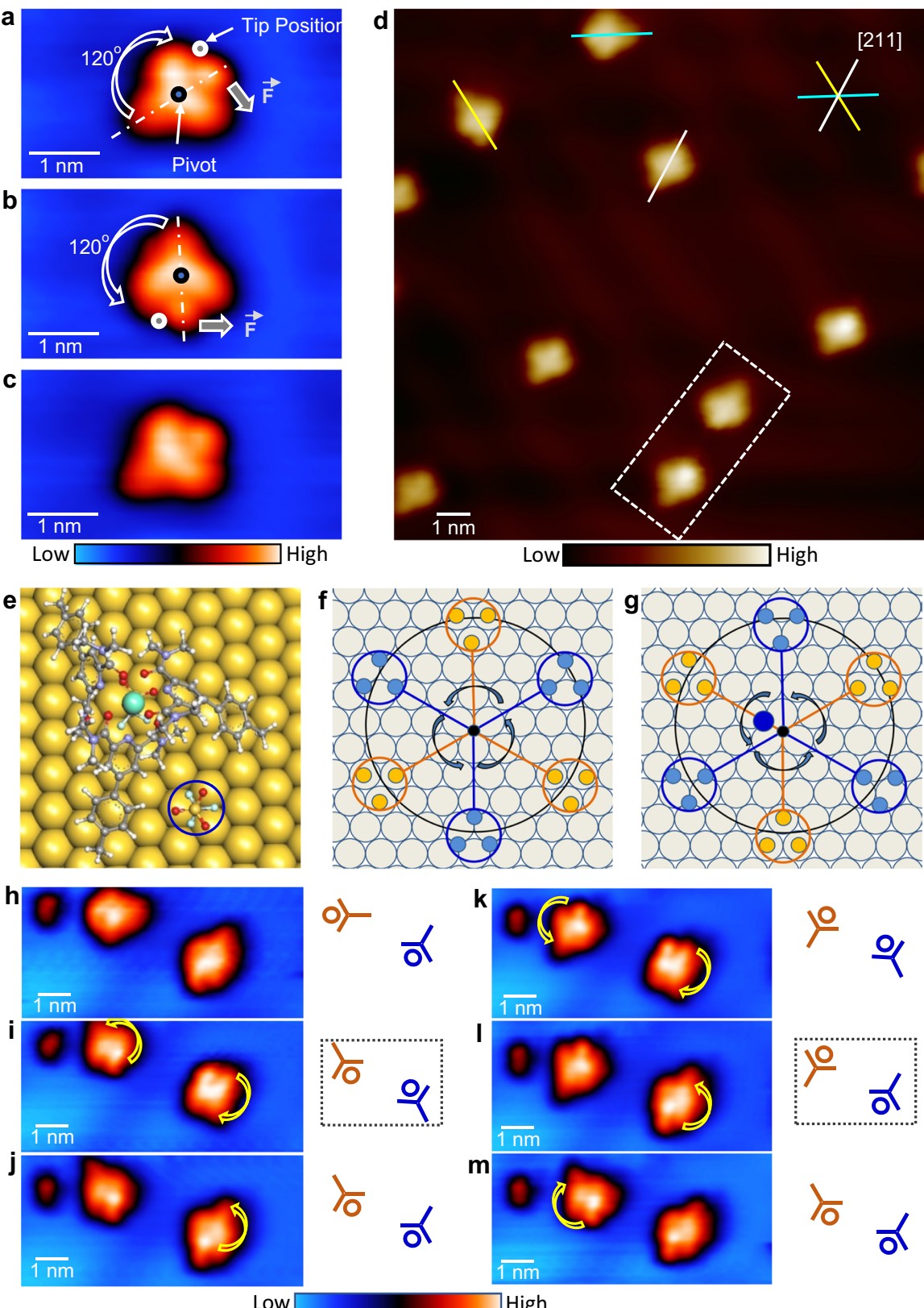

120°; see dashed box in Fig. 5d). This scenario is reminiscent of racemic mixtures of enantiomers, and is solely enforced by the arrangement of fcc and hcp sites on the Au(111) surface (irrespective of the chiral nature of complex [Eu(pcam)₃X₂]⁺). To illustrate this further, a sequence of STM-induced rotations for a pair of complexes [Eu(pcam)₃X₂]⁺ is presented in Fig. 5h–m: the pair of complexes can

never point toward the same direction regardless of the imposed manipulations performed by positioning the tip directly above the centre of the complex (such as shown in Fig. 4i,j). When controlled directional rotations are performed by positioning the STM tip next to the side counterion (such as shown in Fig. 5a–c), this condition is not always prevailed because such manipulation can cause lateral

**Fig. 5 | Atomically precise control of the rotational direction and rotation angle. a–c** A sequence of STM images shows controlled directional rotations. The curved arrows indicate rotation directions and the repulsive force (F) exerted by the negative electric field of the tip to the negatively charged counterion is indicated. **d** An STM image shows scattered complexes [Eu(pcam)$_3$X$_2$]$^+$ pointing along [211] surface directions of Au(111). The white rectangle block highlights a pair of complexes pointing toward opposite directions [V$_t$ = −1.1 V, I$_t$ = 5.8 × 10$^{-11}$A]. **e** Adsorption model of the complex on Au(111). The circle indicates the position of an added counterion. **f, g** Models showing the counterion positions. Here, the

yellow circles indicate adsorption of 3 F atoms on counterion on top surface atom sites while the blue circles are for the adsorption of 3 F atoms on three-fold hollow sites (preferred sites) on Au(111) surface. In (**g**), the complex is shifted for half of Au atomic distance from (**f**) and the adsorption sites of 3 F atoms from the counterion are now reversed. The large blue dot in (**g**) marks the adsorption centre of the complex in (**f**). **h–m** A sequence of STM images and their corresponding models present rotations of two [Eu(pcam)$_3$X$_2$]$^+$ complexes induced by STM tip positioning above the centre of the complex. The dashed boxes in (**i**) and (**l**) indicate when the pair is in opposite directions.

displacement of the complex between fcc and hcp adsorption sites on Au(111) surface.

In summary, we have formed rare-earth complexes on a Au(111) surface by coordinating a positively charged Europium base molecule with negatively charged counterions. The chemical state of the Eu ion in the complexes adsorbed on the surface is determined by synchrotron X-rays STM method where the near-edge X-ray absorption fine structures clearly reveal multiplets originating from the 3+ oxidation state. In accord with this finding, the density functional theory calculations unveil only negligible amount of charge transfer at the molecule-substrate interface, thereby the complexes remained charged on the surface. Formation of charged rare-earth complexes on a metal surface, as demonstrated here, enables investigations on a one-complex-at-a-time basis for their electronic and structural as well as mechanical properties. Furthermore, rotations of the complexes are realized by applying electric field emanating from the STM tip where the complexes rotate using the triflate counterion underneath as a pivot. Using an additional side counterion attached to the complexes and negative electric field of the STM tip we demonstrate 100% directional control over the rotation of these rare-earth complexes. These findings may be useful for the development of nanomechanical devices where the individual units in the complex are designed to control, promote, or restrict motion.

## Methods
### STM experiments
The STM experiments are performed by using two low temperature STM systems, Createc GmbH system at the Center for Nanoscale Materials in Argonne National Laboratory, and a custom-built low temperature STM system in Ohio University. The same results were achieved in both locations ensuring reproducibility. For the measurements, Au(111) single crystal surfaces were cleaned by repeated cycles of Ar or Ne ion sputtering and annealing up to ~700 K under UHV. Then the sample was transferred to the STM chamber in-situ and STM imaging was carried out at ~80 K to assess the cleanliness of the sample surface. Next, the Eu(pcam)$_3$ triflate salt (>99% purity) in a powdered form was deposited onto the atomically clean sample via thermal deposition under UHV. The sample temperature was held at ~200 K during the deposition. The sample was then transferred to the STM scanner under UHV directly attached to the preparation chamber via a gate valve. The experiments were performed at 5 K, 80 K and 100 K substrate temperatures, respectively. Tunneling spectroscopy (dI/dV) and spectroscopy maps were acquired at 5 K substrate temperature. A small lock-in voltage modulation of 20 mV at 1 kHz frequency was used for the spectroscopic measurements.

### SX-STM measurements
SX-STM X-ray absorption spectroscopy (STM-XAS) measurements were performed at ~90 K substrate temperature using the XTIP beamline[30] at the sector 4-ID-E of the Advanced Photon Source and the Center for Nanoscale Materials at Argonne National Laboratory. The X-ray beam is chopped into ON and OFF cycles using an X-ray optical chopper. Then with the aid of the vertical and horizontal focusing mirrors, a final monochromatic X-ray

beam of ~10 μm × 10 μm is delivered onto the tip-sample junction of the SX-STM at the end station. In our SX-STM set-up, a coaxially coated tip is used as a detector to collect the photoexcited current. During the measurements, the detector tip was positioned ~5 nm above the surface and therefore no electron tunneling is taken place. Typically, <1% of the photo-ejected electrons from the sample are collected by the tip, with the photocurrent mostly generated from <1 μm$^2$ area. For the first sample, the Eu(pcam)$_3$ triflate salt in its powdered form was directly pasted onto a freshly cleaved HOPG substrate via mechanical contact. For the second sample, a gold thin-film was grown on a mica substrate, and a sub-monolayer coverage of the the Eu(pcam)$_3$ triflate salt was deposited from solution via drop casting. The sample was annealed at 140 °C for 60 min to evaporate any remaining solvent prior to measurements.

### DFT calculations
The Density Functional Theory (DFT) calculations were performed using the Vienna ab initio simulation package code[36–39], and the core electrons were described by the projected augmented wave method[40]. Exchange-correlation was treated in the Generalized Gradient Approximation, as implemented by ref. [41]. The plane wave basis was expanded to a cut-off of 600 eV and the Brillouin zone was sampled using Γ points only. Calculations are performed for both gas phase and the complexes adsorbed on Au(111) surface. Geometrically relaxed calculations are performed by placing the complexes on a 3 layer Au(111) slab representing the Au(111) surface. Because of the relative importance of non-bonding complex surface interactions, the van der Waals D3 functional was used[42]. Geometrical relaxation is terminated when the change of the total energy is smaller than 0.0001 eV between two ionic steps.

### Synthesis
Starting materials were purchased from Combi-Blocks (San Diego, CA), Sigma-Aldrich (St. Louis, MO), TCI America (Portland, OR), Fisher Scientific (Fair lawn, NJ), AK scientific (Union City, CA), Oakwood Chemical (Estill, SC), Strem Chemicals (Newburyport, MA) and Cambridge Isotope Laboratories (Andover, MA). 1D and 2D spectra were collected using Bruker Ascend 500 MHz and Ultrashield 300 MHz spectrometers at 298 K. CD$_3$CN was used as the solvent; chemical shifts refer to the residual solvent signal of acetonitrile. HR-ESI( + )-MS spectra were collected using a Thermo Fisher Scientific Q Exactive Plus hybrid quadrupole−Orbitrap mass spectrometer in the positive ion mode.

4-Bromo-$N^2$,$N^2$,$N^6$,$N^6$-tetramethylpyridine-2,6-dicarboxamide (**2**)[43] (Fig. 1). A mixture of chelidamic acid **1** (1.0 g, 5.5 mmol) and phosphorus pentabromide (5.9 g, 14 mmol) was heated to reflux at 90 °C under nitrogen for 4 h and cooled to room temperature. The black residue (4-bromopyridine-2,6-dicarbonyl dibromide) was used in the next step without further purification and was dissolved in anhydrous dichloromethane (25 mL) at 0 °C under nitrogen. Anhydrous triethylamine (6.8 mL, 49 mmol) and dimethylamine (2.0 M in tetrahydrofuran, 11 mL, 22 mmol) were added dropwise within 2 min, and the mixture was kept at 0 °C for 20 min, then at room temperature for 2 h. The reaction mixture was filtered through celite with

dichloromethane (10 mL) and concentrated under reduced pressure. The residual black oil was partitioned between water (0.10 L) and dichloromethane (3 × 0.10 L). The combined organic layers were dried over anhydrous sodium sulfate, filtered, and concentrated under reduced pressure to afford a black solid. TLC (ethyl acetate/methanol 10:1): $R_f$ = 0.55. Purification by silica gel chromatography (ethyl acetate/methanol + 1% ammonium hydroxide, 35:1 to 25:1) afforded title compound **2** as a white solid (0.99 g, 3.3 mmol); yield: 60%. $^1$H NMR (500 MHz, CD$_3$CN) $\delta$ 7.76 (s, 2H), 3.04 (s, 6H), 2.95 (s, 6H). $^{13}$C NMR (126 MHz, CD$_3$CN) $\delta$ 167.6 (2), 155.9 (2), 135.0, 127.4 (2), 39.1 (2), 35.5 (2). m.p. 150.8–151.4 °C. HR-MS (ESI) m/z calcd. for C$_{11}$H$_{14}$BrN$_3$O$_2$ ([M + Na]$^+$): 322.01616, found 322.01599.

$N^2,N^2,N^6,N^6$-tetramethyl-4-($p$-tolyl)pyridine-2,6-dicarboxamide (**3**)[44,45] (Fig. 1). A mixture of carboxamide **2** (0.74 g, 2.5 mmol), $p$-tolylboronic acid (0.37 g, 2.7 mmol), tetrakis(triphenylphosphine) palladium (57 mg, 49 μmol), and potassium carbonate (1.0 g, 7.2 mmol) in 1,2-dimethoxyethane (12 mL) and H$_2$O (3.0 mL) was heated to reflux under nitrogen at 90 °C for 48 h. The reaction was cooled to room temperature, brine (50 mL) was added, and the mixture was extracted with ethyl acetate (3 × 50 mL). The combined organic layers were dried over anhydrous sodium sulfate, filtered, and concentrated under reduced pressure to afford a yellow oil. TLC (ethyl acetate/methanol 10:1): $R_f$ = 0.47. Purification by silica gel chromatography (ethyl acetate/methanol + 1% ammonium hydroxide, 40:1, 20:1, 15:1) afforded compound **3** as a colorless oil (0.74 g, 2.4 mmol); yield: 96%. $^1$H NMR (500 MHz, CD$_3$CN) $\delta$ 7.79 (s, 2H), 7.69 (d, $J$ = 8.3 Hz, 2H), 7.35 (d, $J$ = 7.8 Hz, 2H), 3.07 (s, 6H), 2.98 (s, 6H), 2.40 (s, 3H). $^{13}$C NMR (126 MHz, CD$_3$CN) $\delta$ 169.2 (2), 155.7 (2), 151.0, 141.2, 134.9, 130.9 (2), 128.0 (2), 121.2 (2), 39.2 (2), 35.4 (2), 21.3. HR-MS (ESI) m/z calcd. for C$_{18}$H$_{21}$N$_3$O$_2$ ([M + H]$^+$): 312.17065, found 312.17050.

Eu(pcam)$_3$(CF$_3$SO$_3$)$_3$ (**4**) (Fig. 1). A solution of ligand **3** (85 mg, 0.27 mmol) and Europium(III) trifluoromethane sulfonate (55 mg, 92 μmol) in acetonitrile (4.0 mL) was concentrated under reduced pressure to afford complex **4** as a white solid (0.14 g, 91 μmol); yield: 99%. $^1$H NMR (500 MHz, CD$_3$CN) $\delta$ 7.26 (d, $J$ = 8.0 Hz, 2H), 7.21 (d, $J$ = 8.0 Hz, 2H), 6.21 (s, 2H), 3.49 (s, 6H), 3.22 (s, 6H), 2.39 (s, 3H). $^{13}$C NMR (126 MHz, CD$_3$CN) $\delta$ 164.6 (2), 162.7, 144.9 (2), 143.2, 130.7 (2), 130.3, 129.5 (2), 121.9 (CF$_3$), 94.4 (2), 37.5 (2), 36.8 (2), 21.0. $^{19}$F NMR (471 MHz, CD$_3$CN) $\delta$ -79.3. HR-MS (ESI) m/z calcd. for Eu$^{3+}$(C$_{18}$H$_{21}$N$_3$O$_2$)$_3$(SO$_3$CF$_3$)$^-_3$: ([M − (SO$_3$CF$_3$)$^-_3$]$^{3+}$): 362.13657, found 362.13675; ([M − (SO$_3$CF$_3$)$^-_2$]$^{2+}$): 617.68114, found 617.68250.

### Reporting summary
Further information on research design is available in the Nature Research Reporting Summary linked to this article.

## Data availability
All data needed to evaluate the results and conclusions of this study are present within the paper and its Supplementary Materials. Source data are provided with this paper.

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

## Acknowledgements

We acknowledge financial support from the U.S. Department of Energy, Office of Science, Office of Basic Energy Sciences, Materials Science and Engineering Division. Work performed at the Center for Nanoscale Materials, and Advanced Photon Source, both U.S. Department of Energy Office of Science User Facility, was supported by the U.S. DOE, Office of Basic Energy Sciences, under Contract No. DE-AC02-06CH11357. We gratefully acknowledge the computing resources provided on Bebop, a high-performance computing cluster operated by the Laboratory Computing Resource Center at the Argonne National Laboratory.

## Author contributions

S.W.H. lead the project, and conceived and designed the experiments; E.M. designed the ligands and the rare-earth complexes; T.M.A., K.Z.L., S.S., S.P., and S.W. performed STM experiments; T.M.A., S.S., S.W.H, K.Z.L, and S.P. analyzed the STM data; X.C. and F.M. synthesized the complexes; V.J., N.K.D., L.A.C and A.T.N performed the DFT calculations; S.W., N.S., and V.R. performed the synchrotron X-rays experiments. All the authors discussed the results and commented on the paper.

## Competing interests

The authors declare no competing interests.
