## [Peer Review File · Nature Communications]

Atomically Precise Control of Rotational Dynamics in Charged Rare-Earth Complexes on a Metal SurfaceREVIEWER COMMENTS

Reviewer #1 (Remarks to the Author):

This paper reports a 5K STM study of a rare-earth coordination complex that when adsorbed on a Au(111) surface functions as a single molecule rotor and can be driven to rotate unidirectionally via the electric field between the tip and sample. These are very difficult experiments and the data are of excellent quality and in parts very quantitative which is laudable compared to other scanning probe experiments.

The authors appear to have done a good job checking the different surface species observed after the deposition and provide evidence that the molecules with and without the Eu center have different sizes. The STS spectra are a good addition, but are not conclusive for identification of the molecule, but the XS-STM result comparing the oxidation state of the surface bound complexes to "bulk" is excellent and convincing that the Eu stays in a 3+ oxidation state due to minimal charge transfer with the surface, consistent with their DFT calculations.

I am strongly in support of publication in Nature Communications once the following minor points are addressed.

Minor:

Panel of Fig 2 b and c should have same x axis scale for easy comparison. I get why the dI/dV stops at 2.5V but this can still be scaled

Page 8 – "caused by either rotation or switching back-and forth" this is confusing - what is the difference between rotation and switching? Reword

Page 8 – "This highlights that these complexes can function as molecular motors when energized." NO – thermal rotation infers it can function as a rotor when energized, it does not infer anything about the direction of rotation that would make it a "motor"

Page 11 – "It is known that fluorine prefers to adsorb at the three-fold hollow sites on Au(111)²⁸," This is a weak argument as there is a big difference between a F atom and a CF₃ group in terms of preferred binding site of the F which has a different hybridization in each case and could prefer a different site ex an oxygen atom (bridge or three fold) vs a water molecule (atop).

Page 11 "This is an unexpected finding because the latter complex has a larger mass due to additional counterion attached to the side. Thus, it should require a higher critical electric field for rotation." Its nothing to do with mass, its dipole and dispersion interactions that define the barriers. This should be clarified

Reviewer #2 (Remarks to the Author):

Authors report an interesting study on surface-confined rare-earth (RE) complexes that feature rotational dynamics controllable by the tip of a scanning tunneling microscope, as demonstrated by low-temperature measurements on Au substrates.

The subject studied is overall timely, and the work is featuring a very nice complementarity of chemical synthesis, nanoscience experimental techniques and computational modeling.

Scope and content might become suitable for ncomms following careful revision.

Specific comments and criticisms are as follow:

p.1:

The main prospect is ascribed to 'counterionic control', but this aspect is not rigorously treated in the study (lack of systematics etc.) and presumably other implications can be worked out.

Most so-called molecular motors are multicomponent units, as such the reported behaviour is not that surprising upon external stimulation if surface anchoring is weak.

p. 2-3:

Quite a few rotatable RE complexes at surfaces have been reported beforehand, beyond the work cited others studies should be mentioned in this context, whereby also the role of the RE oxidation state has been invoked (e.g., ACS Nano 2009, 3, 1042; Angew. Chem. Int. Ed. 2011, 50, 3872, ACS Nano 2011, 5, 12, 9575; J. Am. Chem. Soc. 2021, 143, 14581 among others). Additionally, the results reported in Phys. Rev. Lett. 101, 197209 are related and relevant.

Last par. (p.2): the assignment of a '2nd, non-coord. counterion at the side of the complex' seems speculative without careful control experiments, such as the deliberate deposition of counterions and their variation.

Also the 'counterion location underneath' should be clearly confirmed, for instance by STM manipulation experiments. Relying exclusively on DFT modeling is not convincing.

Spatially resolved identification of MOs and comparison of STM (dI/dV mapping) data with image simulations and electronic structure would be very helpful to unambiguously confirm adsorption configurations.

p.4:

The STM-XAS approach described is very elegant. Is there an estimate how many adsorbed molecules actually do contribute to the shown signatures?

p.5-6:

It is unclear why 60° rotational angles should be expected just from the presence of the fcc(111) surface atomic lattice symmetry. In principle, if the rotational axis is perpendicular to the surface and defined by the RE center, the nature of the coordination node is decisive.

Rotor and stator elements of the configuration must be clearly designated. If we're dealing with simple rotations of complex adsorbed molecules, the term 'molecular motor' is certainly inappropriate and misleading.

Moreover, assessing the different threshold voltages primarily in terms of the molecular mass is potentially erroneous as different oxidation states of RE centers in adsorbed molecules will sensibly affect surface bonding.

What are the statistics of rotational motion studies? How many of the respective species were probed? Are there variations of results depending on the specific state of the STM tip?

Reviewer #3 (Remarks to the Author):

The authors present a rare-earth complex deposited on a Au(111) surface and show that this complex can rotate when subjected to the electric field in the tunnel junction of a scanning tunnelling microscope (STM). Noteworthy results are the deposition and imaging of the complexes themselves and the in-situ x-ray measurements coupled to STM measurements. I agree with the authors chemical interpretation of the two different species found on the surface as well as the preferred 60° and 120° rotations for [Eu(pcam)3X]2+ [Eu(pcam)3X]2+, respectively. Also, while surprising that such complexes would hold charge while adsorbed on a metal surface (usually quenching is observed), the authors' dI/dV and STM-XAS data is convincing in this regard.

There are a number of claims made in the paper which I am unclear about or dispute. They are listed and described below:

The main disputed point is the claim of 100% control over the rotation direction. To me, this means that the complex can be rotated clockwise or anticlockwise in a deterministic manner. However, the authors show in Figure 3l that, for both of the complexes studied, 46% / 54% or 52% / 48% ratios of clockwise to anticlockwise rotations are observed, respectively. Since the authors give no error on these measurements, one cannot say whether the degree of directionality is statistically significant. In any case, the results are far from a ratio of 100% / 0% as claimed in the paper and abstract.

An additional claim in the abstract is “electric field polarity dependent rotations”. Apart from Fig 3c where two steps in the current (purportedly corresponding to rotation events) are shown at both voltage polarities, no data on the direction dependence of rotation on the voltage polarity is given.

Furthermore, the authors apparently show in Figure 4a-c only two events of “controlled directional rotation”. However, these figures are not discussed (or even mentioned) anywhere in the main text. In the case where this was overlooked by the authors, it is my belief that two isolated events are not enough to support the claim of 100% directionality.

Other points:

- 1) The synthesis and characterisation (supp info pages 2-15) of the $\text{Eu}(\text{pcam})_3$ complex is wholly outside my area of expertise and I therefore do not comment on its validity.
- 2) The authors claim that DFT predicts a counterion located underneath the complex (page 4). The data on the reduction of side-length due to the presence of this counterion is convincing. However, it appears these calculations were done for the gas phase complex rather than the surface bound complex. Could the authors comment on the comparability of these results? Perhaps further evidence could be obtained by laterally manipulating the complex on the surface and revealing a counterion left over on the surface.
- 3) The DFT calculations were performed using VASP but the convergence criteria for their optimised structures is missing and should be added.
- 4) On page 7 the authors compare a bulk powdered sample on HOPG to a submonolayer complex on Au(111). In Fig 2f arbitrary units are given. Should it not be possible to give units here by the current measured during the collection of photoelectrons by the STM tip?
- 5) Fig 3a – it may be better to show more than just one thermally activated freely rotating complex (show a large-scale image). Further, could the authors comment on the onset temperature of the thermally activated rotation? Additionally, the authors state this was observed at a temperature of ~100 K, however, in the methods section it is stated that experiments were performed at 80 K and 5 K.
- 6) Page 10- “Figure 3d to 3e present a sample sequence ofclockwise and anticlockwise directions”, “statistical analysis of rotations...Fig 3f”. These, and other, references to the elements of figure 3 are nonsensical and muddled.
- 7) Fig 3l – it is not stated which pie chart corresponds to which complex.
- 8) The use of the word “energized” in various places in the manuscript is not recommended. This term is rather ambiguous and it is not clear what physical process is at play (for example, statements like “the charges held on the complex interact with the electric field in the junction to cause rotation..” or “energy is imparted though inelastic tunnelling of an electron...” would be more specific and descriptive).
- 9) The use of the term “motor”. As already alluded to, the directionality has not been proved and a motor, by definition, shows directional motion. If no directional rotation is present then it would better be described as a rotor.
- 10) The statement “Such 120-degree rotation angle on a surface with a hexagonal atomic lattice is unprecedented” is not true. 120° rotation has been shown in a number of surface-adsorbed molecules including: B.C. Stipe et al. Science, 279 (1998) 1907; S. Stolz et al, PNAS, 117 (2020) 14838-14842.
- 11) Many STM images (e.g.) appear with a distorted (curved) surface in the background. The surface should be atomically flat. This distortion comes from the line-by-line filter applied to the image and can also affect the interpretation of the molecule’s topography. Better to just do a plane subtraction.

In summary, while an interesting molecular-complex/surface system capable of rotations has been

presented, the data fall short of demonstrating precise control of its rotational dynamics. I do not recommend publication in Nature Communications until the points above have been addressed.

ANSWERS TO REVIEWER COMMENTS

Reviewer #1 (Remarks to the Author): This paper reports a 5K STM study of a rare-earth coordination complex that when adsorbed on a Au(111) surface functions as a single molecule rotor and can be driven to rotate unidirectionally via the electric field between the tip and sample. These are very difficult experiments and the data are of excellent quality and in parts very quantitative which is laudable compared to other scanning probe experiments. The authors appear to have done a good job checking the different surface species observed after the deposition and provide evidence that the molecules with and without the Eu center have different sizes. The STS spectra are a good addition, but are not conclusive for identification of the molecule, but the XS-STM result comparing the oxidation state of the surface bound complexes to “bulk” is excellent and convincing that the Eu stays in a 3+ oxidation state due to minimal charge transfer with the surface, consistent with their DFT calculations. I am strongly in support of publication in Nature Communications once the following minor points are addressed.

ANS: We thank the referee for the constructive comments. Please find our point-by-point answers below.

Minor: Panel of Fig 2b and c should have same x axis scale for easy comparison. I get why the dI/dV stops at 2.5V but this can still be scaled.

ANS: The ‘x’ axes of Fig. 2b and 2c have been rescaled. Moreover, we have now replaced Fig. 2b with a new point dI/dV spectrum with a longer voltage range (-2.87 to +3 V). Longer range dI/dV measurements are difficult because the molecules rotate at higher biases, but we are able to record the data using a retracted tip position.

Page 8 – “caused by either rotation or switching back-and forth” this is confusing - what is the difference between rotation and switching? Reword.

ANS: This sentence is replaced with “random rotations” in page 8.

Page 8 – “This highlights that these complexes can function as molecular motors when energized.” NO – thermal rotation infers it can function as a rotor when energized, it does not infer anything about the direction of rotation that would make it a “motor”.

ANS: We agree with the referee. We have changed the sentence as: “... these complexes can function as molecular motors if their rotation direction could be controlled” in page 8.

Page 11 – “It is known that fluorine prefers to adsorb at the three-fold hollow sites on Au(111)28, ” This is a weak argument as there is a big difference between a F atom and a CF₃ group in terms of preferred binding site of the F which has a different hybridization in each case and could prefer a different site ex an oxygen atom (bridge or three fold) vs a water molecule (atop).

ANS: We have carried out calculations on the entire complex including the side anion for the top and hollow sites of the F atoms positions. The text is now clarified as “for the complex adsorbed

on Au(111) surface The energetic barrier difference between the triflate fluorine atoms” in page 13.

Page 11 “This is an unexpected finding because the latter complex has a larger mass due to additional counterion attached to the side. Thus, it should require a higher critical electric field for rotation.” Its nothing to do with mass, its dipole and dispersion interactions that define the barriers. This should be clarified.

ANS: We have clarified the narrative by adding the following sentence in page 11, “*However, if there is an additional dipole in complex $[Eu(pcam)_3X_2]^+$, it can change the barrier and consequently alter the critical electric field strength required for rotation.*”

During submission of the original manuscript, we mistakenly deleted an entire paragraph related to the explanation of this effect. This paragraph is now added in Page 11 (highlighted).

Reviewer #2 (Remarks to the Author):

Authors report an interesting study on surface-confined rare-earth (RE) complexes that feature rotational dynamics controllable by the tip of a scanning tunneling microscope, as demonstrated by low-temperature measurements on Au substrates. The subject studied is overall timely, and the work is featuring a very nice complementarity of chemical synthesis, nanoscience experimental techniques and computational modeling. Scope and content might become suitable for ncomms following careful revision. Specific comments and criticisms are as follow;

ANS: We thank the referee for the constructive comments. Please find our point-by-point answers below.

p.1: The main prospect is ascribed to ‘counterionic control’, but this aspect is not rigorously treated in the study (lack of systematics etc.) and presumably other implications can be worked out. Most so-called molecular motors are multicomponent units, as such the reported behaviour is not that surprising upon external stimulation if surface anchoring is weak.

ANS: Most molecular motors are multicomponent units however these units are chemically bonded together to form a single molecule. In our case, the system is kept intact by electrostatic interactions, not covalent bonds. To clarify this, the following sentence is added in page 8 (last line): “*multiple components held together just by electrostatic interactions*”.

We have performed additional experiments and calculations comprehensively for the counterionic control. The new results demonstrate compelling evidence of the counterionic control. Specifically, statistical analysis from 709 rotations (523 rotations from 8 sequences are provided in supplementary information S9) clearly shows a deterministic control of the rotation directions. We have also explained the mechanism of controlled rotation in the text (in page 11) and in supplementary information (S8).

p. 2-3: Quite a few rotatable RE complexes at surfaces have been reported beforehand, beyond the work cited others studies should be mentioned in this context, whereby also the role of the RE oxidation state has been invoked (e.g., ACS Nano 2009, 3, 1042; Angew. Chem. Int. Ed. 2011, 50, 3872, ACS Nano 2011, 5, 12, 9575; J. Am. Chem. Soc. 2021, 143, 14581 among others). Additionally, the results reported in Phys. Rev. Lett. 101, 197209 are related and relevant.

ANS: We thank the referee for suggesting these relevant references. These references are now added (ref. 23-27).

Last par. (p.2): the assignment of a '2nd, non-coord. counterion at the side of the complex' seems speculative without careful control experiments, such as the deliberate deposition of counterions and their variation.

ANS: The triflate (CF_3SO_3^-) is negatively charged. Therefore, the sample composed of pure counterion cannot be formed. Likewise, $\text{Eu}(\text{pcam})_3$ cannot exist as a separate charged species without counterions. The source sample is a neutral salt – its purity is above 99%. Moreover, our calculations reveal that attaching the counterion to the centre of the $[\text{Eu}(\text{pcam})_3]^{3+}$ unit is most favorable, while attaching it to the sides is the second-best alternative in the gas phase as well. Furthermore, the counterion is extensively proved by STM manipulations where its negative charge is the key for the controlled directional rotations as discussed in page 11, and supplementary information S8 and S9.

Also the 'counterion location underneath' should be clearly confirmed, for instance by STM manipulation experiments. Relying exclusively on DFT modeling is not convincing. Spatially resolved identification of MOs and comparison of STM (dI/dV mapping) data with image simulations and electronic structure would be very helpful to unambiguously confirm adsorption configurations.

ANS: In this revised version, we have made a clear and unequivocal distinction between the complexes with and without counterion underneath using a pair of $[\text{Eu}(\text{pcam})_3\text{X}]^{2+}$ - $[\text{Eu}(\text{pcam})_3]^{3+}$ complexes.

(1) Incorporation of the counterion underneath changes the electronic structure and orbital shapes drastically between the complexes with and without counterions. This provides a compelling evidence of counterion incorporation. The spectroscopic maps agree very well with the theory calculations (added in Supplementary Information S2, S4, and S5).

(2). $[\text{Eu}(\text{pcam})_3]^{3+}$ has a symmetric triangular shape with the side length of 2.1 nm while $[\text{Eu}(\text{pcam})_3\text{X}]^{2+}$ is bent and distorted, which reduces the side arm length to 1.8 nm. The comparison between the side lengths of the complexes provides the expected values.

We have also added a dI/dV spectroscopic movie of this complex pair, "Diff-Mol-Orbitals-Movie", in supplementary information, which is created from 8000 spectroscopic frames. The spectroscopic maps are acquired at every 1 mV from +2000 mV to -2000 mV and, to ensure reproducibility, repeated back to +2000 mV. This movie was acquired simultaneously with Fig.

S24a, and it clearly reveals the differences in orbital shapes between 1200 mV and 2000 mV where the LUMOs of complexes are located.

Because of a strong electrostatic interaction between the positively charged $\text{Eu}(\text{pcam})_3$ and negatively charged counterion, they cannot be separated just by STM manipulation. This always results in movement of the entire complex because the binding to the surface is weaker than the binding between the counterion and the $\text{Eu}(\text{pcam})_3$ unit.

p.4: The STM-XAS approach described is very elegant. Is there an estimate how many adsorbed molecules actually do contribute to the shown signatures?

ANS: Typically, less than 1% of the illuminated area is recorded by the STM-XAS at the tip channel. The measurements were performed at ~ 5 nm tip height (not in tunneling regime). Considering $10 \mu\text{m} \times 10 \mu\text{m}$ X-ray beam size, the complexes from an area of $< 1 \mu\text{m}^2$ can contribute the signals (up to $\sim 10^5$ molecules). The following sentences are added in the “Methods” section in page 15 and 16;

“In our SX-STM set-up, a coaxially coated tip is used as a detector to collect the photoexcited current. During the measurements, the detector tip was positioned ~ 5 nm above the surface and therefore no electron tunneling is taken place. Typically, less than 1% of the photo-ejected electrons from the sample are collected by the tip, with the photocurrent mostly generated from $< 1 \mu\text{m}^2$ area.”

p.5-6: It is unclear why 60° rotational angles should be expected just from the presence of the fcc(111) surface atomic lattice symmetry. In principle, if the rotational axis is perpendicular to the surface and defined by the RE center, the nature of the coordination node is decisive.

ANS: The rotational plane of the $\text{Eu}(\text{pcam})_3$ is parallel to the surface and therefore it is expected to follow the periodic potential barrier imposed by the hexagonal symmetry of the atomic lattice.

Rotor and stator elements of the configuration must be clearly designated. If we’re dealing with simple rotations of complex adsorbed molecules, the term ‘molecular motor’ is certainly inappropriate and misleading.

ANS: The rotor is the $\text{Eu}(\text{pcam})_3$ unit while the stator is the counterion underneath the molecule. This is now clearly defined in the text as, *“the pivot here is the triflate counterion underneath the $\text{Eu}(\text{pcam})_3$ unit (Fig. 1b), which acts as a stator while the $\text{Eu}(\text{pcam})_3$ unit acts as a rotator.”* (Added in page 10). The rotations discussed in Fig. 3 can be considered as rotors however controlled directional rotations discussed in Fig. 4 should be considered as motors.

Moreover, assessing the different threshold voltages primarily in terms of the molecular mass is potentially erroneous as different oxidation states of RE centers in adsorbed molecules will sensibly affect surface bonding.

ANS: We agree with the referee about the mass. We have removed the sentence.

What are the statistics of rotational motion studies? How many of the respective species were probed? Are there variations of results depending on the specific state of the STM tip?

ANS: Statistical data were collected from a total of 1049 tip induced rotations, out of which, 340 rotations belong to Fig. 3 while the remaining 709 rotations are for the tip induced controlled rotations. Rotations are induced by stepwise one rotation event at-a-time and each rotation event is confirmed by taking STM images before and after the rotation. STM images of 8 rotation sequences (523 controlled directional rotations) are added in supplementary (Fig. S31 to Fig. S53). We have also added the statistics in the text (page 10).

In addition, the experiments were performed using 2 LT-STM systems in two separate locations (due to COVID travel restriction) and the exact same results are reproduced. The following sentences are added in Method Section (in page 15); *“The STM experiments are performed by using two low temperature STM systems, Createc GmbH system in Argonne National Laboratory, and a custom-built low temperature STM system in Ohio University. The same results were achieved in both locations ensuring reproducibility.”*

Reviewer #3 (Remarks to the Author):

The authors present a rare-earth complex deposited on a Au(111) surface and show that this complex can rotate when subjected to the electric field in the tunnel junction of a scanning tunnelling microscope (STM). Noteworthy results are the deposition and imaging of the complexes themselves and the in-situ x-ray measurements coupled to STM measurements. I agree with the authors chemical interpretation of the two different species found on the surface as well as the preferred 60° and 120° rotations for [Eu(pcam)3X]2+ [Eu(pcam)3X2]+, respectively. Also, while surprising that such complexes would hold charge while adsorbed on a metal surface (usually quenching is observed), the authors' dI/dV and STM-XAS data is convincing in this regard. There are a number of claims made in the paper which I am unclear about or dispute. They are listed and described below:

ANS: We thank the referee for the constructive comments. Please find our point-by-point answers below.

The main disputed point is the claim of 100% control over the rotation direction. To me, this means that the complex can be rotated clockwise or anticlockwise in a deterministic manner. However, the authors show in Figure 3I that, for both of the complexes studied, 46% / 54% or 52% / 48% ratios of clockwise to anticlockwise rotations are observed, respectively. Since the authors give no error on these measurements, one cannot say whether the degree of directionality is statistically significant. In any case, the results are far from a ratio of 100% / 0% as claimed in the paper and abstract.

ANS: During submission of the original manuscript, we mistakenly deleted a paragraph related to the counterion control. This paragraph is now added in Page 11.

When the tip is positioned directly above the centre of the molecule, then it can rotate both clockwise and anticlockwise (the images and discussions related to Fig. 3). Here, we do not have a control over the rotational direction.

The following sentence is added in page 10; “... *thus no directional control of their rotations is achieved*”.

However, when the tip is positioned next to the negative counter ion, then the rotational direction can be precisely controlled (Fig. 4a to 4c, and Supplementary Information S8, and S9). Here, the Coulomb repulsion between negatively charged counterion and the negative electric field of the STM tip always results in pushing the complex from one end, which leads to the rotation using the counterion underneath the complex as a pivot. Therefore, depending on the tip position either left or right side of the side counterion, both clockwise and anticlockwise rotation can be induced in a deterministic manner.

We have added STM image frames of 8 rotation sequences (523 rotations) in supplementary information S9. We also explain the rotation mechanism in the text (added in page 11), as well as in the supplementary information S8.

An additional claim in the abstract is “electric field polarity dependent rotations”. Apart from Fig 3c where two steps in the current (purportedly corresponding to rotation events) are shown at both voltage polarities, no data on the direction dependence of rotation on the voltage polarity is given.

ANS: As mentioned above, both polarities can induce rotation when the tip is positioned above the centre of the complex but there is no directional control (materials related to Fig. 3). We can control the rotational direction only when the tip is positioned next to the counterion (materials related to Fig. 4). The controlled rotation can be done only with the negative electric field of the tip. For the positive electric field, the tip usually picks up the complex through the negatively charged side counterion destroying the imaging condition. This explanation is in the added paragraph in page 11.

Furthermore, the authors apparently show in Figure 4a-c only two events of “controlled directional rotation”. However, these figures are not discussed (or even mentioned) anywhere in the main text. In the case where this was overlooked by the authors, it is my belief that two isolated events are not enough to support the claim of 100% directionality.

ANS: We apologize again for our mistake. As mentioned above, we have now added the paragraph explaining the Fig. 4a to 4c in page 11 (highlighted). Additional statistical analyses on directional rotations are also added in the supplementary information. We have performed 709 controlled rotation events and STM frames of 8 sequences containing 523 rotations are added in supplementary information S9.

Other points:

1) The synthesis and characterisation (supp info pages 2-15) of the $\text{Eu}(\text{pcam})_3$ complex is wholly outside my area of expertise and I therefore do not comment on its validity.

ANS: Since this is a new rare-earth complex system, we have added the synthesis information for the benefit of the synthetic chemistry community.

2) The authors claim that DFT predicts a counterion located underneath the complex (page 4). The data on the reduction of side-length due to the presence of this counterion is convincing. However, it appears these calculations were done for the gas phase complex rather than the surface bound complex. Could the authors comment on the comparability of these results? Perhaps further evidence could be obtained by laterally manipulating the complex on the surface and revealing a counterion left over on the surface.

ANS: The DFT calculations were done for both gas phase (Fig. 1b, and supplementary information) and on Au(111) surface (Fig. 1c, 2c, 2d, and 2e and supplementary information Fig S21a, Fig. S27b and d). In this revised version, we have made a clear and unequivocal distinction between the complexes with and without counterion underneath using a pair of $[\text{Eu}(\text{pcam})_3\text{X}]^{2+}$ - $[\text{Eu}(\text{pcam})_3]^{3+}$ complexes.

(1) Incorporation of the counterion underneath changes the electronic structure and orbital shapes drastically between the complexes with and without counterions. This provides a compelling evidence of counterion incorporation. The spectroscopic maps agree very well with the theory calculations (added in Supplementary Information S2, S4, and S5).

(2). $[\text{Eu}(\text{pcam})_3]^{3+}$ has a symmetric triangular shape with the side length of 2.1 nm while $[\text{Eu}(\text{pcam})_3\text{X}]^{2+}$ is bent and distorted, which reduces the side arm length to 1.8 nm. The comparison between the side lengths of the complexes provides the expected values.

We have also added a dI/dV spectroscopic movie of this complex pair, "Diff-Mol-Orbitals-Movie", in supplementary information, which is created from 8000 spectroscopic frames. The spectroscopic maps are acquired at every 1 mV from +2000 mV to -2000 mV and, to ensure reproducibility, repeated back to +2000 mV. This movie was acquired simultaneously with Fig. S24a, and it clearly reveals the differences in orbital shapes between 1200 mV and 2000 mV where the LUMOs of complexes are located.

Because of a strong electrostatic interaction between the positively charged $\text{Eu}(\text{pcam})_3$ and negatively charged counterion, they cannot be separated just by STM manipulation. This always results in movement of the entire complex because the binding to the surface is weaker than binding between the counterion and the $\text{Eu}(\text{pcam})_3$ unit.

3) The DFT calculations were performed using VASP but the convergence criteria for their optimized structures is missing and should be added.

ANS: The following sentence is added in the method section (page 16).

“Geometrical relaxation is terminated when the change of the total energy is smaller than 0.0001 eV between two ionic steps.”

4) On page 7 the authors compare a bulk powdered sample on HOPG to a sub-monolayer complex on Au(111). In Fig 2f arbitrary units are given. Should it not be possible to give units here by the current measured during the collection of photoelectrons by the STM tip?

ANS: We have changed the arbitrary unit to measured photocurrent values in Fig. 2f.

5) Fig 3a – it may be better to show more than just one thermally activated freely rotating complex (show a large-scale image). Further, could the authors comment on the onset temperature of the thermally activated rotation? Additionally, the authors state this was observed at a temperature of ~100 K, however, in the methods section it is stated that experiments were performed at 80 K and 5 K.

ANS: We have provided a largescale image that shows multiple rotating complexes in supplementary information (S10). The complexes do not rotate at 80 K (~ LN₂ temperature). They rotate at ~100 K. We initially found this effect by chance while checking deposition conditions in one of the two LT-STM systems used. The results were reproduceable.

We have added “100 K” in the method section (in page 15).

6) Page 10- “Figure 3d to 3e present a sample sequence ofclockwise and anticlockwise directions”, “statistical analysis of rotations...Fig 3f”. These, and other, references to the elements of figure 3 are nonsensical and muddled.

ANS: Statistical data were collected from a total of 1049 tip induced rotations, out of which, 340 rotations belong to Fig. 3 while the remaining 709 rotations are for the tip induced controlled rotations. STM frames of 8 rotation sequences (523 controlled rotations) are added in supplementary.

7) Fig 3l – it is not stated which pie chart corresponds to which complex.

ANS: We have added clarification in the figure 3l caption as “type A and B complexes”. The pi charts are labeled ‘A’ and ‘B’.

8) The use of the word “energized” in various places in the manuscript is not recommended. This term is rather ambiguous and it is not clear what physical process is at play (for example, statements like “the charges held on the complex interact with the electric field in the junction to cause rotation..” or “energy is imparted though inelastic tunnelling of an electron...” would be more specific and descriptive).

ANS: The word “energized” is removed throughout the text.

9) The use of the term “motor”. As already alluded to, the directionality has not been proved and a motor, by definition, shows directional motion. If no directional rotation is present then it would better be described as a rotor.

ANS: We have now added compelling evidence of the directional control in supplementary information. We have also added 2 STM movies (stacked frames) showing clockwise and anticlockwise rotations. Explanations on the controlled rotation mechanism are added in page 11 and in supplementary information (S8, and S9).

10) The statement “Such 120-degree rotation angle on a surface with a hexagonal atomic lattice is unprecedented” is not true. 120° rotation has been shown in a number of surface-adsorbed molecules including: B.C. Stipe et al. Science, 279 (1998) 1907; S. Stolz et al, PNAS, 117 (2020) 14838-14842.

ANS: We are grateful to the referee for mentioning these two important references. These references are now added as reference 33, and 34, and the word “unprecedented” is removed.

11) Many STM images (e.g.) appear with a distorted (curved) surface in the background. The surface should be atomically flat. This distortion comes from the line-by-line filter applied to the image and can also affect the interpretation of the molecule’s topography. Better to just do a plane subtraction.

ANS: We have corrected the STM images in all the figures.

In summary, while an interesting molecular-complex/surface system capable of rotations has been presented, the data fall short of demonstrating precise control of its rotational dynamics. I do not recommend publication in Nature Communications until the points above have been addressed.

REVIEWERS' COMMENTS

Reviewer #2 (Remarks to the Author):

Revision was successful.

However, 2 answers on p. 4 of the rebuttal seem contradictory:

p.5-6: It is unclear why 60° rotational angles should be expected just from the presence of the fcc(111) surface atomic lattice symmetry. In principle, if the rotational axis is perpendicular to the surface and defined by the RE center, the nature of the coordination node is decisive.

ANS: The rotational plane of the Eu(pcsm)₃ is parallel to the surface and therefore it is expected to follow the periodic potential barrier imposed by the hexagonal symmetry of the atomic lattice.

Rotor and stator elements of the configuration must be clearly designated. If we're dealing with simple rotations of complex adsorbed molecules, the term 'molecular motor' is certainly inappropriate and misleading.

ANS: The rotor is the Eu(pcsm)₃ unit while the stator is the counterion underneath the molecule. This is now clearly defined in the text as, "the pivot here is the triflate counterion underneath the Eu(pcsm)₃ unit (Fig. 1b), which acts as a stator while the Eu(pcsm)₃ unit acts as a rotor." (Added in page 10). The rotations discussed in Fig. 3 can be considered as rotors however controlled directional rotations discussed in Fig. 4 should be considered as motors.

Reviewer #3 (Remarks to the Author):

After reading the authors revised manuscript as well as their response to my comments on the originally submitted article, I feel that they have addressed my points in a satisfactory manner. Thanks to the supplementary videos (as well as the substantial (!) data set added in the supp info), I now have no doubt that the complex can be rotated in a directional manner.

The only remaining point of contention I have is the use of "...complexes act as single molecule motors..." in the abstract. The authors nicely show that the direction of rotation can be controlled by selecting the tip position ("Depending on the tip position left or right side of the side counterion, clockwise or anticlockwise rotation can be performed at will"). For a motor I think, strictly speaking, the direction of rotation should be intrinsic to the system and should not depend on the STM user choosing to apply electrostatic repulsion on one side of the molecule or the other. To give two opposing examples: ref 34 has directionality imposed by chirality and rotates in one direction regardless of the manner of excitation and thus may be termed a motor; whereas ref 22 shows control over the direction of rotation but this is very much dependent on which way the molecule is "pushed" using the field in the STM junction and was therefore not presented as a motor.

In general, the work is interesting and I therefore now recommend publication in Nature Communications.

Kind regards,
Reviewer 3

REVIEWERS' COMMENTS

Reviewer #2 (Remarks to the Author):

Revision was successful.

However, 2 answers on p. 4 of the rebuttal seem contradictory: p.5-6: It is unclear why 60° rotational angles should be expected just from the presence of the fcc(111) surface atomic lattice symmetry. In principle, if the rotational axis is perpendicular to the surface and defined by the RE center, the nature of the coordination node is decisive.

ANS: The rotational plane of the Eu(pcam)₃ is parallel to the surface and therefore it is expected to follow the periodic potential barrier imposed by the hexagonal symmetry of the atomic lattice.

Rotor and stator elements of the configuration must be clearly designated. If we're dealing with simple rotations of complex adsorbed molecules, the term 'molecular motor' is certainly inappropriate and misleading.

ANS: The rotor is the Eu(pcam)₃ unit while the stator is the counterion underneath the molecule. This is now clearly defined in the text as, "the pivot here is the triflate counterion underneath the Eu(pcam)₃ unit (Fig.1b), which acts as a stator while the Eu(pcam)₃ unit acts as a rotator." (Added in page 10). The rotations discussed in Fig. 3 can be considered as rotors however controlled directional rotations discussed in Fig. 4 should be considered as motors.

ANS: It is not contradictory. However, to avoid confusion, we have removed the sentence, "...60° rotational angles should be expected".

Reviewer #3 (Remarks to the Author):

After reading the authors revised manuscript as well as their response to my comments on the originally submitted article, I feel that they have addressed my points in a satisfactory manner. Thanks to the supplementary videos (as well as the substantial (!) data set added in the supp info), I now have no doubt that the complex can be rotated in a directional manner.

The only remaining point of contention I have is the use of "...complexes act as single molecule motors..." in the abstract. The authors nicely show that the direction of rotation can be controlled by selecting the tip position ("Depending on the tip position left or right side of the side counterion, clockwise or anticlockwise rotation can be performed at will"). For a motor I think, strictly speaking, the direction of rotation should be intrinsic to the system and should not depend on the STM user choosing to apply electrostatic repulsion on one side of the molecule or the other. To give two opposing examples: ref 34 has directionality imposed by chirality and rotates in one direction regardless of the manner of excitation and thus may be termed a motor; whereas ref 22 shows control over the direction of rotation but this is very much dependent on

which way the molecule is "pushed" using the field in the STM junction and was therefore not presented as a motor.

ANS: We have replaced the sentence "...complexes act as single motors" with "the entire complex rotates as a single unit" in the abstract. Direct labeling of the complex as a motor is removed in the main text and in the supplementary information file.